# Ki-67 is a PP1-interacting protein that organises the mitotic chromosome periphery

Daniel G Booth[1], Masatoshi Takagi[2], Luis Sanchez-Pulido[3], Elizabeth Petfalski[1], Giulia Vargiu[1], Kumiko Samejima[1], Naoko Imamoto[2], Chris P Ponting[3], David Tollervey[1], William C Earnshaw[1]*, Paola Vagnarelli[4]*

[1]Wellcome Trust Centre for Cell Biology, University of Edinburgh, Edinburgh, United Kingdom; [2]Cellular Dynamics Laboratory, Riken Advanced Science Institute, Wako Saitama, Japan; [3]MRC Functional Genomics Unit, Department of Physiology, Anatomy and Genetics, University of Oxford, Oxford, United Kingdom; [4]Biosciences, Brunel University, London, United Kingdom

**Abstract** When the nucleolus disassembles during open mitosis, many nucleolar proteins and RNAs associate with chromosomes, establishing a perichromosomal compartment coating the chromosome periphery. At present nothing is known about the function of this poorly characterised compartment. In this study, we report that the nucleolar protein Ki-67 is required for the assembly of the perichromosomal compartment in human cells. Ki-67 is a cell-cycle regulated protein phosphatase 1-binding protein that is involved in phospho-regulation of the nucleolar protein B23/nucleophosmin. Following siRNA depletion of Ki-67, NIFK, B23, nucleolin, and four novel chromosome periphery proteins all fail to associate with the periphery of human chromosomes. Correlative light and electron microscopy (CLEM) images suggest a near-complete loss of the entire perichromosomal compartment. Mitotic chromosome condensation and intrinsic structure appear normal in the absence of the perichromosomal compartment but significant differences in nucleolar reassembly and nuclear organisation are observed in post-mitotic cells.

*For correspondence: bill.
earnshaw@ed.ac.uk (WCE);
Paola.Vagnarelli@brunel.ac.uk
(PV)

Reviewing editor: Anthony A
Hyman, Max Planck Institute of
Molecular Cell Biology and
Genetics, Germany

## Introduction

When mitotic chromosomes are examined by whole mount microscopy the surface chromatin is obscured by a layer of proteins and RNA derived from the dense fibrillar component (DFC) and granular component (GC) of the nucleolus (*Moyne and Garrido, 1976*; *Harrison et al., 1982*; *Adolph and Kreisman, 1983*; *Sumner, 1991*; *Gautier et al., 1992b*; *Wanner et al., 2005*). This perichromosomal layer includes pre-rRNA, U3 snoRNAs, and over 20 ribosomal proteins, including nucleolin and Nopp140, NPM/B23, Bop1, Nop52, PM-Scl 100, and Ki-67 (*Gautier et al., 1992a, 1994*; *Hernandez-Verdun and Gautier, 1994*; *Van Hooser et al., 2005*; *Fomproix et al., 1998*; *Angelier et al., 2005*). The perichromosomal layer represents 1.4% of the chromosome proteome (*Ohta et al., 2010*). At present, its functional significance remains unstudied (*Van Hooser et al., 2005*).

Ki-67 is one of the earliest proteins to bind the perichromosomal layer in mitosis. Ki-67 is an antigen recognised by a monoclonal antibody generated by immunizing mice with nuclei of Hodgkin lymphoma cells (*Gerdes et al., 1983*). Because Ki-67 is nuclear only in proliferating cells, it is widely used as a marker to assess cell proliferation. For example, immuno-histochemical assessment of the proportion of cells staining for nuclear Ki-67 is used to predict the responsiveness or resistance of tumours to therapy (*Dowsett et al., 2011*).

**eLife digest** The genetic information of an organism is found in the nucleus of each cell in the form of DNA organised into chromosomes. The exact structure of those chromosomes changes as the cell moves through the different stages of the cell division cycle. During the stage called mitosis, where the DNA of a cell (which has previously been duplicated) is shared into two daughter cells, the chromosomes become tightly packed structures that can be readily moved through the cytoplasm. Since the late nineteenth century, it has been known that a layer of proteins, called the perichromosomal layer, coats the condensed chromosomes. However, virtually nothing was known about the role this layer performs.

One of the first proteins to join the perichromosomal layer after mitosis begins is called Ki-67. This is only found in the cell nucleus when a cell is actively growing and dividing, and so is widely used as a marker in experiments investigating these processes: for example, Ki-67 is used to detect growing tumour cells amongst the normal cells in tissues of the body, and to measure the effectiveness of drugs designed to stop the growth of tumours. Again, however, little is known about what Ki-67 actually does.

Booth et al. now reveal that when Ki-67 is not present in a cell, chromosomes do not have a perichromosomal layer—or at best, have a small remnant of one. This allowed Booth et al. to investigate the role of the perichromosomal layer as well. When the chromosomes first go through mitosis without a perichromosomal layer, no changes to the shape or the behaviour of the chromosomes are seen. However, the new nuclei are smaller than normal and their contents are arranged differently. This causes problems with the ability of daughter cells to synthesise protein building blocks and leads to an increased rate of spontaneous cell death when daughter cells try to undergo the next mitosis. Further research is needed to understand why this happens.

Ki-67 protein is predominantly localised in the cortex and dense fibrillar components of the nucleolus during interphase (*Kill, 1996*; *MacCallum and Hall, 2000b*). During mitosis it relocates to the periphery of the condensed chromosomes (*Verheijen et al., 1989b*; *Isola et al., 1990*). It has been reported that Ki-67 is associated with the nuclear matrix (*Verheijen et al., 1989b*), preribosomes (*Isola et al., 1990*), satellite DNA in G1 (*Bridger et al., 1998*), and with the chromosome scaffold of mitotic cells (*Verheijen et al., 1989a*). Approximately 40% of the cellular pool of protein is present on isolated mitotic chromosomes (*Ohta et al., 2010*).

Previous studies suggested that Ki-67 is regulated by phosphorylation. Hyperphosphorylated Ki-67 does not bind DNA during mitosis (*Endl and Gerdes, 2000*; *MacCallum and Hall, 1999*) and is characterized by increased mobility, on mitotic chromosomes measured by FRAP. In interphase, non-phosphorylated Ki-67 binds DNA and does not recover after FRAP (*Saiwaki et al., 2005*). Its localisation and cell-cycle behaviour suggest that Ki-67 could be involved in the organisation of nucleolar chromatin during interphase in proliferating cells. Recently, it was reported that Ki-67 also functions in mitosis in stabilisation and maintenance of the mitotic spindle by recruiting Hklp2 to mitotic chromosomes (*Vanneste et al., 2009*).

In this study, we have identified Ki-67 as ancestral to the PP1 targeting subunit Repo-Man (*Trinkle-Mulcahy et al., 2006*) and have shown that Ki-67 is a PIP (PP1 Interacting Protein) that contributes to the phospho-regulation of nucleophosmin/B23 by CKII. Ki-67 is also a major organiser of the perichromosomal layer, possibly acting as an interaction platform. Remarkably, Ki-67 depletion prevents all nucleolar proteins tested from accumulating around the chromosomes in mitosis. Thus, chromosomes in cells depleted of Ki-67 appear to lack most or all of their perichromosomal layer. This enabled us to gain insights into the function of this enigmatic chromosomal compartment. Interestingly, loss of the perichromosomal layer does not detectibly compromise the condensed morphology or intrinsic structure of mitotic chromosomes but does result in significant changes in nucleolar morphology and nuclear organization in the following interphase.

## Results

### Ki-67 is a PP1-interacting protein in vivo

We used a phylogenetic approach to identify putative functional regions within the sequence of Repo-Man, a targeting protein that binds PP1 in a cell-cycle specific manner regulated by a phospho-switch

(*Trinkle-Mulcahy et al., 2006*; *Qian et al., 2011*; *Vagnarelli et al., 2011*). A BLAST search revealed significant ($E = 5 \times 10^{-4}$) similarity between a small region (amino acids 388–420) of human Repo-Man and Ki-67 (*Figure 1A1,2*), a very large protein that exhibits strong links to cell proliferation (*Gerdes et al., 1983*). The region conserved between Repo-Man and Ki-67 contains the PP1 binding motif (RVTF) of Repo-Man, which is conserved as RVSF in human Ki-67 (*Figure 1C3*).

The similarity between Repo-Man and Ki-67 within the PP1 binding domain suggested that Ki-67 could be a PP1-interacting protein (PIP). Indeed, a possible connection between Ki-67 and PP1 was previously identified in two large scale screens for PP1 interactors. In one, in silico-screening based on a stringent definition of the RVxF motif (the PP1 binding motif) identified Ki-67 as putative inter-actor, however no in vivo interaction was demonstrated (*Hendrickx et al., 2009*). Another study identified Ki-67 in a displacement affinity chromatography analysis of PP1 nuclear interactors (*Moorhead et al., 2008*).

To analyse if Ki-67 was a cell-cycle regulated PIP in vivo, we performed a tethering/recruitment experiment (*Vagnarelli et al., 2011*). Ki-67$^{301–700}$ wild type (wt) and a PP1 non-binding RASA mutant were fused to GFP:Lac repressor (*Figure 1B*) and co-expressed with RFP:PP1γ or β in DT40 cells carrying a lacO array integrated on a single chromosome (*Vagnarelli et al., 2011*). GFP:Lac repressor on its own was used as a control in this experiment.

All GFP:Lac repressor fusion constructs accumulated at the LacO integration site (*Figure 1C1′–3′*). Furthermore, the Ki-67 wt construct recruited PP1 to the same spot (*Figure 1C2″*). Quantification of the PP1 signal intensity at the GFP:Lac repressor spot compared to the general nuclear distribution (excluding nucleoli) revealed that Ki-67 recruitment of PP1 is more efficient in interphase than in mitosis and that PP1γ is recruited more efficiently than PP1β during interphase (*Figure 1D*). These data suggest that Ki-67 is a PIP in vivo and the interaction with PP1γ is cell-cycle regulated.

In order to determine whether Ki-67 is required to target PP1γ to any of its known locations in vivo, we used RNAi to deplete Ki-67 in human cells and determined the effects that this had on the distribution of PP1γ. A previous report described the successful depletion of Ki-67 (*Vanneste et al., 2009*). However, the target sequence recognised by the published siRNA lies within a repeated stretch and rescue experiments to validate the phenotype were not reported. We therefore identified two new siRNAs that could deplete the protein as efficiently as the published siRNA (*Figure 1*, *Figure 1—figure supplement 1*). Both new siRNAs yielded comparable phenotypes and we used siRNA 5 (Ki5) for the depletion experiments presented below. This siRNA was validated in a rescue experiment that is discussed in a later section (*Figure 2B*).

Ki-67 depletion in a HeLa cell line has no effect on the accumulation of RFP:PP1γ in the nucleolus (*Figure 1*, *Figure 1—figure supplement 2[1,4]*). Indeed, the targeting subunit for PP1 nucleolar locali-sation has been recently reported to be RRP1B (*Chamousset et al., 2010*). In early mitosis, PP1γ localised normally on the spindle and at kinetochores in both control and Ki-67 depleted cells (*Figure 1*, *Figure 1—figure supplement 2[2,5]*). However, we observed a significant decrease in PP1 levels on anaphase chromatin in Ki-67 depleted cells (*Figure 1*, *Figure 1—figure supplement 2[3′,6′]*). Previous reports identified Repo-Man and Sds22 as responsible for targeting PP1 to anaphase chromatin (*Trinkle-Mulcahy et al., 2006*; *Wurzenberger et al., 2013*). Thus, Ki-67 is one of the several factors contributing to the accumulation of PP1γ on chromatin during mitotic exit.

## Ki-67 regulates B23 phosphorylation

Analysis of the phosphorylation status of several known direct and indirect Ki-67 interacting proteins (*Figure 1E*) in interphase and mitosis revealed that nucleophosmin/B23 phospho-regulation was dependent on Ki-67. B23 is phosphorylated both in interphase and in mitosis by several kinases (*Pfaff and Anderer, 1988*; *Jiang et al., 2000*; *Louvet et al., 2006*; *Krause and Hoffmann, 2010*; *Ramos-Echazabal et al., 2012*; *Reboutier et al., 2012*), including CyclinB/CDK1 at T199 (*Tokuyama et al., 2001*) in mitosis and by casein kinase II (CKII) on S125 during interphase (*Szebeni et al., 2003*).

Use of phospho-specific antibodies revealed a reproducible difference in nucleophosmin/B23 phosphorylation on S125 in the presence and absence of Ki-67 exponential cultures and in prometaphase cells (*Figure 1F*). In both cases, the levels of S125ph were significantly increased following Ki-67 depletion. This was particularly evident in prometaphase-arrested cells. In contrast, we observed no significant difference in the phosphorylation status of B23 at T199 in the presence or absence of Ki-67 (data not shown). These experiments support the notion that Ki-67 is a functional PP1-targeting sub-unit in vivo.

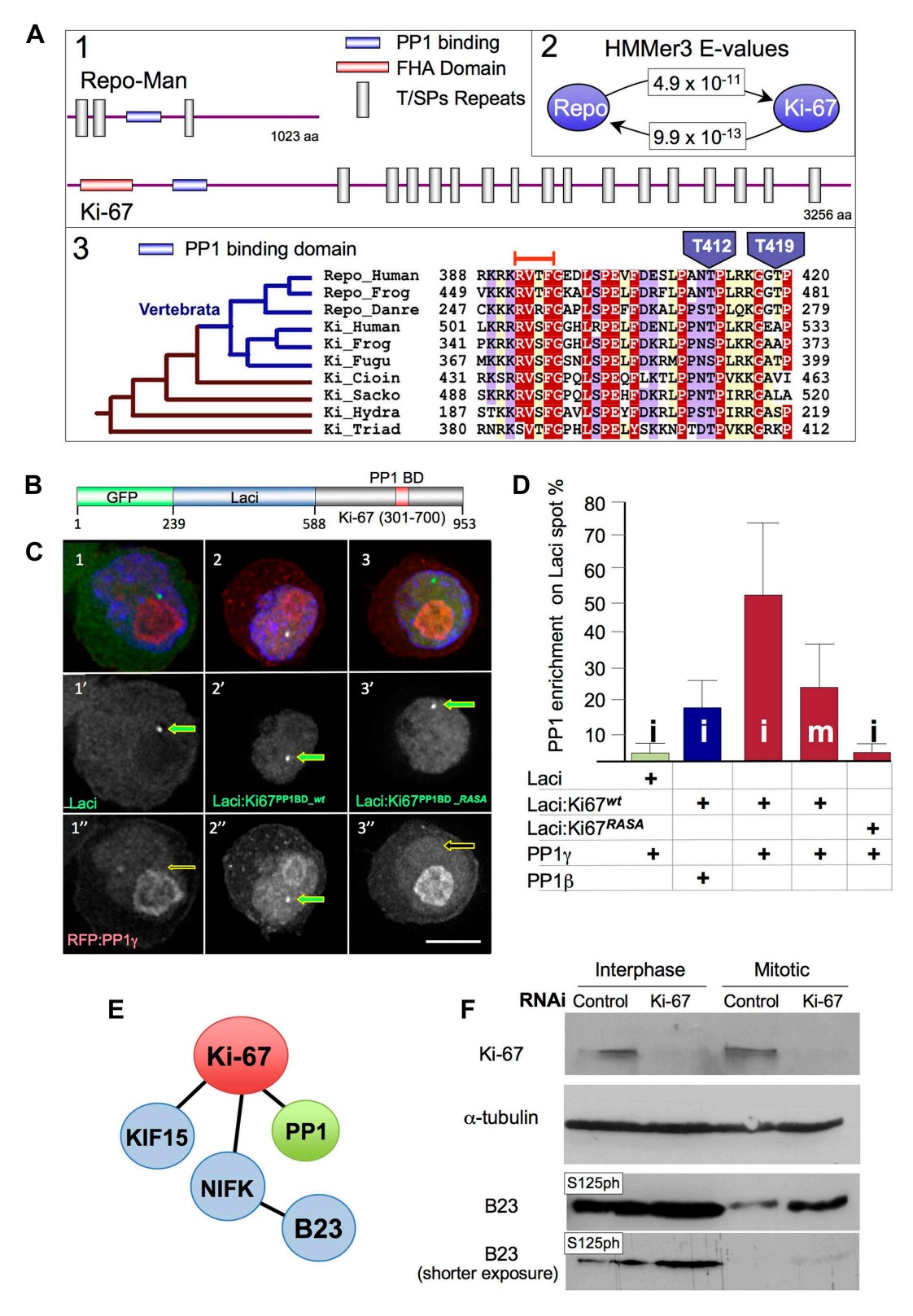

**Figure 1**. Ki-67 is evolutionary related to Repo-Man but shows distinct behaviour during mitosis. (**A1**) Schematic representations of evolutionarily conserved regions in human Repo-Man and Ki-67 proteins (shown approximately to scale). (**A2**) *E*-values corresponding to global profile-to-sequence (HMMer3) comparisons between the PP1 binding conserved regions (blue oval) in Repo-Man and Ki-67 families. Arrows indicate the profile search

*Figure 1. Continued on next page*

*Figure 1. Continued*

direction. The Repo-Man profile identified Ki-67 proteins as homologous with a highly significant *E*-value of $4.9 \times 10^{-11}$. Conversely, the profile of Ki-67 homologous sequences in animals identified Repo-Man proteins with a significant *E*-value of $9.9 \times 10^{-13}$. (**A3**) Representative multiple sequence alignment of conserved regions from Repo-Man and Ki-67 families. Important mitotic phospho-residues in Repo-Man (T412 and T419) are indicated in blue. The RVTF motif is indicated with an orange box. The most parsimonious explanation of Repo-Man and Ki-67 evolution is shown to the left of the alignment. Vertebrate branches are coloured in blue. Sequences are named according to: Repo_Human, NCBI:NP_689775, *Homo sapiens*; Repo_Frog, NCBI:ACR33033, *Xenopus laevis*; Repo_Danre, UniProt:A2CEF0, *Danio rerio*; Ki_Human, UniProt:P46013, *Homo sapiens*; Ki_Frog, UniProt:Q0VA85, *Xenopus laevis*; Ki_Fugu, UniProt:UPI00016EA029, *Takifugu rubripes*; Ki_Cioin, UniProt:UPI000180CFDA, *Ciona intestinalis*; Ki_Sacko, Baylor College of Medicine genome and FGENESH+, *Saccoglossus kowalevskii*; Ki_Hydra, UniProt:UPI0001926DD5, *Hydra magnipapillata*; and, Ki_Triad, UniProt:B3SB24, *Trichoplax adhaerens*. The amino acid colouring scheme indicates average BLOSUM62 scores (which are correlated with amino acid conservation) for each alignment column: red (greater than 3.5), violet (between 3.5 and 2.5), and light yellow (between 2.5 and 0.5). (**B–C**) Ki-67 and PP1γ interact in vivo. Ki-67$^{301–700}$ fused to Lac repressor:GFP (Lac repressor:Ki-67$^{PP1BD\_wt}$) was transfected together with RFP:PP1γ into a DT40 cell line containing a LacO array integrated on a single chromosome. Ki-67 was enriched at the LacO site (panels 2, 2′) where it recruited PP1γ (2, 2″). However, neither Lac repressor:GFP (panels 1–1″) or the Ki-67 PP1-non-binding mutant (Lac repressor:Ki-67$^{PP1BD\_RASA}$) (panels 3–3″) caused PP1 accumulation at the LacO site. (**D**) Ki-67 recruits PP1γ more efficiently than PP1beta and more efficiently in interphase than in mitosis. The experimental set up was as in (**B**). The enrichment of PP1 signal at the Lac repressor spot was compared to the background nuclear (interphase) or cytoplasmic (mitosis) signal within the same cell. Scale bar 10 μm. (**E**) Direct and indirect interactors of Ki-67. (**F**) The B23$^{S125}$ phosphorylation level remains high in Ki-67 depleted mitotic cells. Immunoblots of whole cell extracts of cycling (interphase) of Nocodazole-arrested (mitotic) HeLa cells transfected with Ki-67 RNAi oligo 5 or control oligos, were probed for Ki-67, tubulin, B23$^{T199ph}$, and B23$^{S125ph}$. Two exposures of the B23$^{S125ph}$ blot are shown.

The following figure supplements are available for figure 1:

**Figure supplement 1**. Characterisation of Ki-67 RNAi.

**Figure supplement 2**. Distribution of PP1gamma in mitosis after the Ki-67 siRNA.

## Lack of Ki-67 compromises the assembly of the perichromosomal compartment in mitosis

Several aspects of mitotic chromosome structure remain relatively poorly understood, but amongst these, the perichromosomal compartment (also known as the chromosome periphery) stands out as a structure about which almost nothing is known. This is remarkable, as an ever-increasing list of chromosome periphery proteins has been compiled over the years (*Chaly et al., 1984*; *McKeon et al., 1984*; *Gautier et al., 1992b*; *Hernandez-Verdun and Gautier, 1994*; *Van Hooser et al., 2005*; *Gassmann et al., 2005*; *Ohta et al., 2010*). Some of these are among the most abundant proteins associated with mitotic chromosomes (*Ohta et al., 2010*). Despite their abundance, the mechanism of their localisation and the role that they play on the chromosomes, if any, is still not understood (*Hernandez-Verdun and Gautier, 1994*; *Van Hooser et al., 2005*). In a recent proteomic study, we identified a number of novel chromosome periphery proteins, which we termed cPERPs (*Ohta et al., 2010*).

Ki-67 is one of many nucleolar proteins that are recruited to the chromosome periphery at the transition from prophase to prometaphase. As Ki-67 is recruited to the perichromosomal compartment relatively early, we decided to examine whether its depletion affected the localisation of other known chromosome periphery proteins.

We first examined the localisation of the endogenous nucleolin using a specific antibody. Indeed, nucleolin failed to accumulate at the chromosome periphery in the absence of Ki-67 (*Figure 2A*). The phenotype was observed using all of the siRNAs that we have tested and was rescued by an siRNA-resistant cDNA (*Figure 2B*, *Figure 2—figure supplements 1, 2*).

To examine whether Ki-67 depletion affected the behaviour of other chromosome periphery proteins, we next examined the behaviour of NIFK, a known KI-67 interactor (*Takagi et al., 2001*). During normal mitotic exit, NIFK concentrates around the segregating chromosomes before associating with nucleolar-derived foci (NDF–*Dundr et al., 1996*; *Figure 2C3*). NDF formation varies between cell lines (*Dundr et al., 2000*), but in the HeLa cells that we have used for RNAi studies, prominent NDFs are seldom observed during unperturbed mitosis (*Figure 2C1–4*).

To follow NIFK behaviour, we transiently expressed GFP-tagged hNIFK in HeLa cells after control or Ki-67 RNAi. The nucleolar localisation of GFP-NIFK in interphase was unaltered after Ki-67 depletion (*Figure 2C8*), but in early mitosis the protein failed to properly accumulate around the mitotic

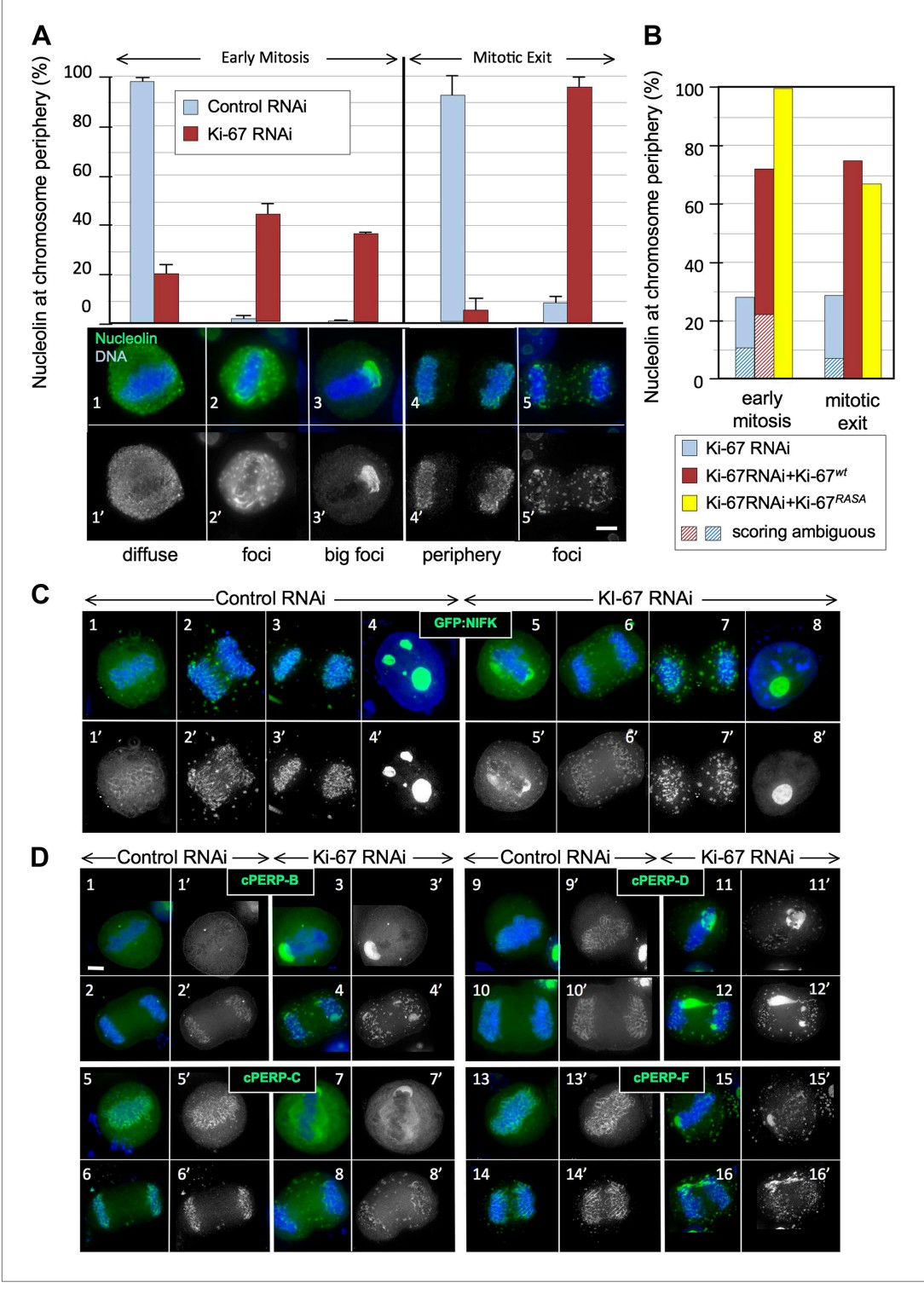

**Figure 2**. Ki-67 is required for targeting of nucleolar proteins to the chromosome periphery in mitosis.
(**A**) Localisation of endogenous nucleolin is aberrant in mitotic cells after Ki-67 depletion (panels 2, 3, 5). HeLa cells were transfected with Ki-67 RNAi oligo 5 (panels 2, 3, 5) or control oligos (panels 1, 4) and stained for nucleolin (green). Quantification of the phenotypes is indicated in the graph above the corresponding representative images. Scale bar 5 µm. (**B**) RNAi rescue experiment. HeLa cells depleted of Ki-67 were transfected with either mCherry:Ki-67$^{wt}$, or mCherry:Ki-67$^{RASA}$, together with Ki-67 RNAi oligo 5 or control oligo and stained for nucleolin. The localisation
*Figure 2. Continued on next page*

*Figure 2. Continued*

of nucleolin in mitotic cells was quantified in the different experimental conditions. See ***Figure 2—figure supplement 1*** for representative images. Scale bar 10 μm. (**C**) The mitotic chromosome peripheral localisation of NIFK is disrupted upon Ki-67 RNAi (panels 5–6). HeLa cells were transfected with GFP:NIFK (green) and oligo 5 (panels 5–8) or control oligo (panels 1–4). Scale bar 10 μm. (**D**) All novel cPERPs tested failed to accumulate on the chromosome periphery in mitosis. HeLa cells were co-transfected with GFP:cPERPs identified in an earlier study (***Ohta et al., 2010***) (green) and oligo 5 (panels 3–4, 7–8, 11–12, 15–16) or control oligo (panels 1–2, 5–6, 9–10, 13–14): GFP:cPERP-B (panels 1–4), GFP:cPERP-C (panels 5–8), GFP:cPERP-D (panels 9–12), GFP:cPERP-F (panels 13–16). Scale bar 5 μm.

The following figure supplements are available for figure 2:

**Figure supplement 1**. Distribution of nucleolin in mitosis following exposure of cells to different Ki-67 siRNA oligonucleotides with and without rescue by Ki-67 cDNA.

**Figure supplement 2**. Distribution of nucleolin in mitosis following exposure of cells to different Ki-67 siRNA oligonucleotides.

**Figure supplement 3**. Distribution of NIFK in mitosis following Ki-67 depletion.

chromosomes (***Figure 2C5–7***). Instead, in metaphase cells, it accumulated in large cytoplasmic aggregates, one of which was frequently found to 'cap' one end of the cluster of chromosomes on the metaphase plate (***Figure 2C5***). This extraordinary behaviour was also seen for endogenous NIFK phosphorylated on Thr234 using a phospho-specific antibody (***Figure 2—figure supplement 3[3']***). During mitotic exit, in the absence of Ki-67, GFP-NIFK accumulated in NDF-like cytoplasmic aggregates that persisted until the reformation of the nuclear membrane (***Figure 2C5–7***).

In view of the striking similarity of the behaviour of nucleolin and NIFK in the absence of Ki-67, we tested the generality of this effect by localizing four novel cPERPs identified in our proteomics studies (***Ohta et al., 2010***). Remarkably, all were mislocalised in Ki-67-depleted cells, and all were frequently observed to 'cap' one end of the metaphase plate (***Figure 2D3,7,11,15***). As in the case of nucleolin and NIFK, these aggregates dispersed into clusters of smaller cytoplasmic foci during mitotic exit (***Figure 2D4,8,12,16***).

We considered two hypotheses to explain the observed failure of nucleolar proteins to associate with the chromosome periphery in mitosis of Ki-67-depleted cells. First, Ki-67 might be required for the complete disassembly of the nucleolus during mitotic entry. In this case, the larger cytoplasmic aggregates observed during metaphase might be remnants of incompletely disassembled nucleoli. Correlative light-microscopy/Electron microscopy (CLEM) analysis of cells transfected with both GFP-cPERP-C and the indicated siRNA oligos eliminated this hypothesis. The GFP-containing aggregates observed in mitosis in Ki-67-depleted cells did not correspond to nucleolar remnants or any other recognisable electron-dense structures such as NDFs (***Dundr et al., 2000***; ***Figure 3A***, ***Figure 3—figure supplement 1***).

A second hypothesis suggested that Ki-67 might function as a scaffolding platform for organising the perichromosomal compartment. Indeed, all candidate proteins that we have tested to date have failed to localise to the chromosomal periphery in Ki-67-depleted cells. Although analyses of individual proteins can provide part of the story, it is difficult or impossible to make conclusions about the compartment as a whole since the chromosome periphery has a complex and as-yet undefined protein and RNA composition. Nonetheless, further analysis of our CLEM images strongly suggests that depletion of Ki-67 causes a loss of most or all of the perichromosomal compartment.

Examination of electron micrographs of control and Ki-67-depleted cells at higher magnification revealed a subtle difference in the appearance of the edge of the chromosomes. In control cells there was a 'fuzzy' transition zone between the electron-dense chromatin and the cytoplasm (***Figure 3B***). This transition appeared to be more abrupt in Ki-67-depleted chromosomes. This was confirmed by line scans across the chromosome-to-cytoplasm boundary, which showed that there is normally a gradual decrease in density at the edge of mitotic chromosomes. This transition was significantly more abrupt in the absence of Ki-67 (***Figure 3C***).

These experiments suggest that Ki-67 directs the binding of nucleolar components to the chromosome periphery, perhaps by acting as a scaffold. Alternatively, the macromolecular network at the

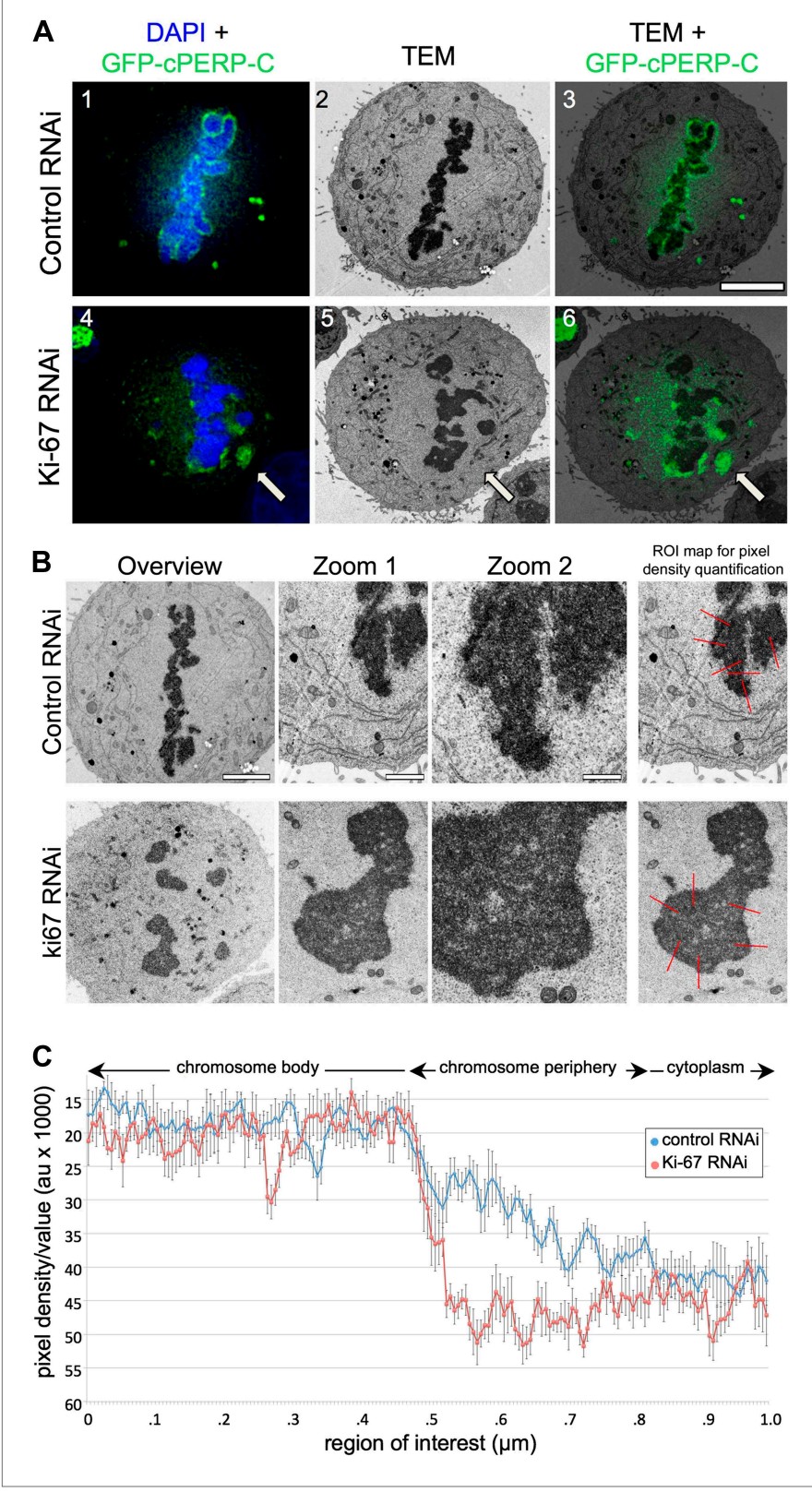

**Figure 3**. Ki-67 organizes the mitotic chromosome periphery. (**A**) CLEM of HeLa cells transfected with GFP-PerP-C and control oligos (top panels) or Ki-67 oligos (bottom panels). Mitotic cells from control or Ki-67 RNAi with visible GFP aggregates (arrow) were identified and processed for CLEM using an adapted protocol (**Booth et al., 2011**). *Figure 3. Continued on next page*

*Figure 3. Continued*

Appropriate light and electron micrographs, from the matching z position, were overlaid using Adobe Photoshop Elements. Scale bar 5 μm. (**B**) EM of mitotic cells from Control (top panels) and Ki-67 RNAi (bottom panels). At higher magnification (zoom 2) it is possible to note that the amorphous material surrounding the mitotic chromosomes in control cells (arrow) is substantially reduced around Ki-67 depleted chromosomes. Scale bars (left to right) 4, 2, and 1 μm. Far right panels show regions of interest for pixel density analysis (*Figure 3C*). (**C**) Line scans of the peripheral regions of mitotic chromosomes in control (blue line) and Ki-67 depleted cells (red line).

The following figure supplements are available for figure 3:

**Figure supplement 1**. Depletion of Ki-67 does not leave electron-dense nucleolar remnants in the cytoplasm.

chromosomal periphery could be delicately balanced, such that removal of any single component causes the entire structure to fail. However, depletion of cPERP-B, -D, or -E using RNAi (*Figure 4—figure supplement 1*) failed to alter the striking perichromosomal localization of Ki-67 (*Figure 4*). This is consistent with Ki-67 being upstream of the other components in the assembly of the perichromosomal compartment.

## Organisation of the perichromosomal layer does not require PP1 binding by Ki-67

The observations presented above suggest two possible hypotheses for how Ki-67 could function in the assembly of the chromosome periphery. First, the protein might comprise part of a PP1 holoenzyme whose removal of phosphates from chromosomal proteins, could be required for those proteins to assemble at the chromosome periphery. Alternatively, Ki-67, which is a very large protein with multiple repeated domains, might function as a scaffold linking together key components at the chromosome periphery. To begin to distinguish between these two hypotheses, we generated constructs expressing mRFP coupled to either wild-type Ki-67 or a Ki-67 RASA mutant that is incapable of binding PP1 at the conserved site shared with Repo-Man (*Figure 1A3*). Both constructs encoded mRNAs that were engineered to be resistant to siRNA-5.

Both mRFP:Ki-67-wt and mRFP:Ki-67-RASA proteins rescued the localisation of endogenous nucleolin throughout mitosis in cells transfected with Ki-67 siRNAs. In transfected cells, nucleolin was once again concentrated at the chromosome periphery from prometaphase through telophase (*Figure 2B*, *Figure 2—figure supplement 1*). When combined with our previous observations these data support the hypothesis that Ki-67 acts as a scaffold for formation of the perichromosomal compartment. Importantly, this rescue experiment also confirmed that the chromosome periphery phenotypes observed following Ki-67 depletion by siRNA-5 are not due to off-target effects.

## Searching for a function of the perichromosomal compartment in mitotic chromosomes

Because depletion of Ki-67 appears to result in either a dramatic reduction, or even complete loss, of the perichromosomal compartment, these experiments offer a unique opportunity to investigate the function(s) of this mysterious structure. If the perichromosomal compartment is a kind of pellicle or 'skin' on the surface of the chromosomes (*Schrader, 1944*), then two obvious potential functions come to mind.

First, the perichromosomal layer could protect the chromosomal DNA from damage once it is released into a potentially hostile cytoplasmic environment following nuclear envelope breakdown. Immunostaining for 53BP1, a protein that associates with DNA damage foci revealed that levels of intrinsic DNA damage are low in mitotic cells under our culture conditions (*Figure 5A,B*). These levels were not substantially increased in chromosomes lacking Ki-67. Thus, this hypothesis appears unlikely.

An alternative hypothesis is that by forming a layer coating each chromosome, the perichromosomal compartment could promote the formation of physically separated chromosomes that is characteristic of mitosis. Our prior studies revealed that condensin, the chromokinesin Kif4 and DNA topoisomerase II all contribute to shaping mitotic chromosomes (*Hudson et al., 2003*; *Green et al., 2012*; *Samejima et al., 2012*). Furthermore, proteomic analysis revealed a substantial reduction in levels of Ki-67 associated with isolated mitotic chromosomes depleted of KIF4A

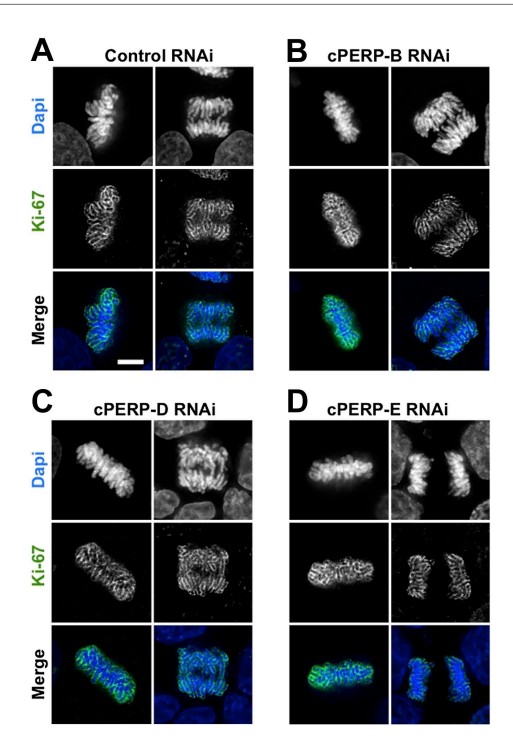

**Figure 4**. Ki-67 localisation is not dependant on other c-PERPS tested. (**A**) Ki-67 localisation was analysed following the RNAi depletion of several novel c-PERPs (PERP **B**, **D,** and **E**). Following a 48 hr knock-down period, cells were fixed and labelled with anti-Ki67 (green) antibody and Dapi (blue). Examples shown include metaphase and anaphase cells. Scale bar 5 µm.
The following figure supplements are available for figure 4:

**Figure supplement 1**. Depletion of cPERPs by RNAi.

(*Samejima et al., 2012*). The converse is not true, however, as neither the localisation nor abundance (as determined by immunostaining) of KIF4A was altered in the absence of Ki-67 (*Figure 5C*). Thus, Ki-67 depletion does not affect the known mitotic chromosome structural proteins.

Mitotic chromosomes have an intrinsic metaphase structure (IMS) that can be probed using a specialized assay, in which chromosomes are induced to unfold down to the level of 10 nm fibres by removal of divalent cations, and then induced to re-fold by re-addition of $Mg^{2+}$ (*Earnshaw and Laemmli, 1983*; *Hudson et al., 2003*). Even though chromosomes lacking condensin or KIF4 appear morphologically normal when cells are processed under optimal conditions, those chromosomes are severely impaired in the IMS assay—failing to re-adopt a normal appearance after restoration of divalent cations (*Hudson et al., 2003*; *Samejima et al., 2012*). Other abundant chromosomal proteins, including, cohesin and DNA topoisomerase II are not required for this aspect of mitotic chromosome structure (*Vagnarelli et al., 2004*; *Johnson et al., 2009*).

We transfected cells with either control or Ki-67 siRNAs and then performed the IMS assay. This experiment showed clearly that chromosomes depleted of Ki-67 efficiently regain their normal morphology after two rounds of unfolding and refolding. Thus, the perichromosomal layer is not required for maintenance of the intrinsic structure of mitotic chromosomes (*Figure 5D,E*).

In summary, we could find no evidence for a role of Ki-67—and by extension much or all of the perichromosomal compartment—in mitotic chromosome structure or integrity under these conditions.

## KI-67 is required for normal segregation of nucleolin

Given the abnormal localization of nucleolar components during mitosis, it is no surprise that the segregation of these components is perturbed in mitosis. The detailed behaviour of the complex population of nucleolar components following abolishment of the perichromosomal compartment is a subject for a follow-up study, but here as an example, we have examined the segregation of the abundant component nucleophosmin/B23.

We followed cell division in living HeLa cells expressing GFP-B23 after either control or Ki-67 siRNA. In Ki-67-depleted cells, B23 was never observed to associate with the chromosomes during mitosis. Instead, it started forming cytoplasmic aggregates just after anaphase onset (*Figure 6A*, 45'). We then analysed the distribution of endogenous nucleolin between daughter cells in cytokinesis. While in control cells there was a relatively even distribution of chromatin-associated nucleolin between daughter cells, in Ki-67-depleted cells the nucleolin aggregates were often not chromatin bound and their distribution between daughter cells tended to be more asymmetric (*Figure 6B,C*).

Despite this perturbation of the behaviour of major nucleolar components during mitosis, nucleoli did indeed re-form during mitotic exit. Furthermore, the ultrastructure of the reformed nucleoli appeared to be normal, with fibrillar centres, dense fibrillar component, and granular component all readily observed (*Figure 6—figure supplement 1*). Thus, Ki-67 is apparently not required for the structural organisation within interphase nucleoli.

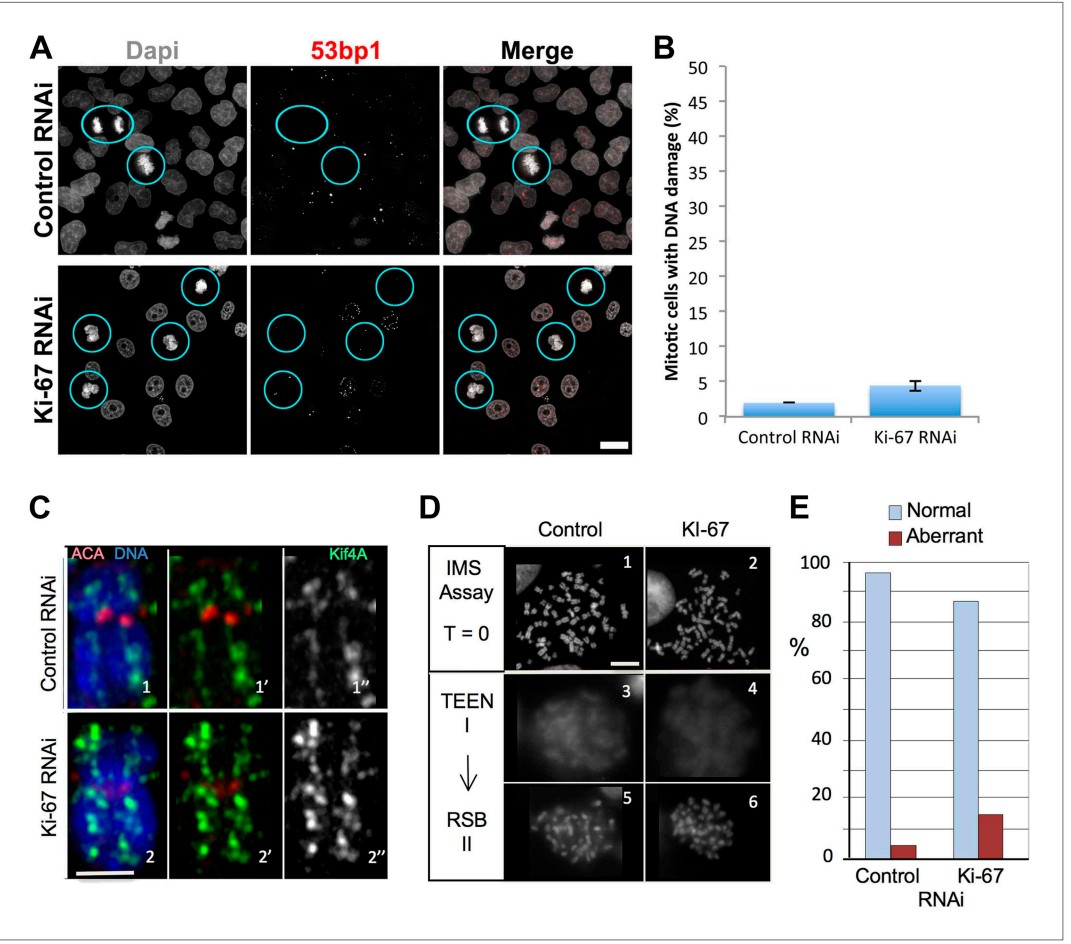

**Figure 5**. Ki-67 is does not function to protect chromosomes from DNA damage or provide structural maintenance. (**A**) Representative overview images of HeLa cells transfected with control or Ki-67 specific siRNA oligos probed with anti-53bp antibodies to assess levels of DNA damage. Scale bar 10 μm. (**B**) A bar graph showing quantification of the percentage of mitotic cells found with DNA damage, marked using an anti-53bp antibody. (**C**) Chromosome spreads from control RNAi (panels 1–1") and Ki-67 oligo 5 RNAi (panels 2–2") were stained for the chromosome scaffold protein KIF4A (green) and for the kinetochore with ACA (red). Ki-67 depleted chromosomes still maintain a proper localisation of the chromosome scaffold and kinetochore proteins. Scale bar 2 μm. (**D**) IMS Assay (Intrinsic Metaphase Structure Assay). Chromosomes from HeLa cells after Ki-67 depletion (panels 2, 4, 6) and control depletion (panels 1, 3, 5) at the beginning of the assay (panels 1, 2), after the first TEEN treatment (panels 3, 4), and after the second RSB addition (panels 5, 6). Scale bar 10 μm. (**E**) Quantification of the experiment in (**D**).

## KI-67 is required for normal reactivation of nucleolar organizer regions following mitotic exit

Human cells contain five chromosomes with ribosomal gene clusters that can form NORs. Thus, human cells could in theory have 10 nucleoli. This is almost never seen, presumably due to natural clustering of the NORs and the failure of all NORs to become activated. In our line of HeLa cells, we observed 3.5 ± 0.3 nucleoli per nucleus (*Figure 7D,E*). This changed dramatically following Ki-67 depletion, when the number of nucleoli observed dropped to 1.6 ± 0.3. This strongly suggests that chromosomes lacking a perichromosomal layer might associate with one another more closely than normal, thus promoting NOR fusion during reactivation.

A second reason why Ki-67-depleted cells might have single large nucleoli could be due to a decreased efficiency of NOR reactivation during mitotic exit. Ki-67-depleted cells tended to be smaller and to have smaller nuclei (*Figure 7A–C*, *Figure 7—figure supplement 1*). This was not due to an accumulation of the cells in G1, as shown by FACs analysis (*Figure 7—figure supplement 2*), but could potentially be due to ribosomal insufficiency.

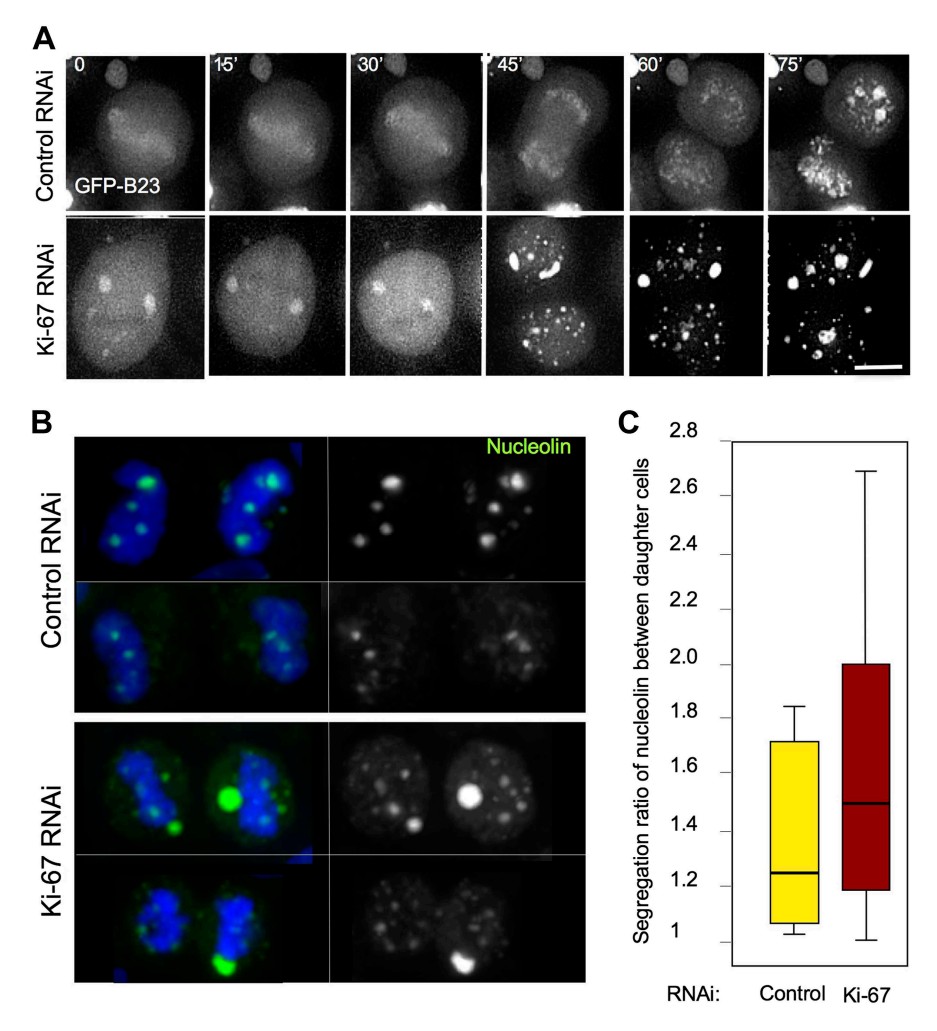

**Figure 6**. Segregation of nucleolin. (**A**) In cells depleted of Ki-67 the nucleolus never disassembles completely during mitosis and B23 never accumulates around the mitotic chromosomes. Time-lapse imaging of GFP:B23 in HeLa cells after control or Ki-67 RNAi. Scale bar 10 µm. (**B**) Nucleolin localisation was analysed in cells in cytokinesis after control or Ki67 RNAi. Nucleolin distribution between the two daughter cells is uneven following Ki-67 RNAi compared to the control and the protein is predominantly not associated with the chromatin. (**C**) Quantification of the experiment in (**B**). The graphs represent the ratio of the total nucleolin intensity between daughter cells.

The following figure supplements are available for figure 6:

**Figure supplement 1**. Electron micrographs of interphase nuclei from Ki-67 RNAi cells.

We have made several observations that are consistent with this. First, if we calculate the total nucleolar area in optical sections of Ki-67-depleted nuclei, we find that these nuclei tend to have a smaller aggregate nucleolar area (*Figure 7F,G*, *Figure 7—figure supplement 3*). If we assume total nucleolar area to be a proxy for the number of genes involved in rDNA transcription, this suggests that reactivation of ribosomal clusters following mitotic exit may be less efficient in Ki-67-depleted cells. This is consistent with the results of a Northern analysis of total rRNA in control and Ki-67-depleted cells, which revealed decreased levels of pre-rRNA species, consistent with lower levels of rRNA transcription in the depleted cells (*Figure 7H*).

To further examine NOR reactivation following mitotic exit, we exploited the fact that chromosomes bearing re-activated NORs are associated with nucleoli. We used a HT1080 cell line carrying a LacO array integrated on chromosome 13p next to the NOR and expressing GFP fused to Lac repressor

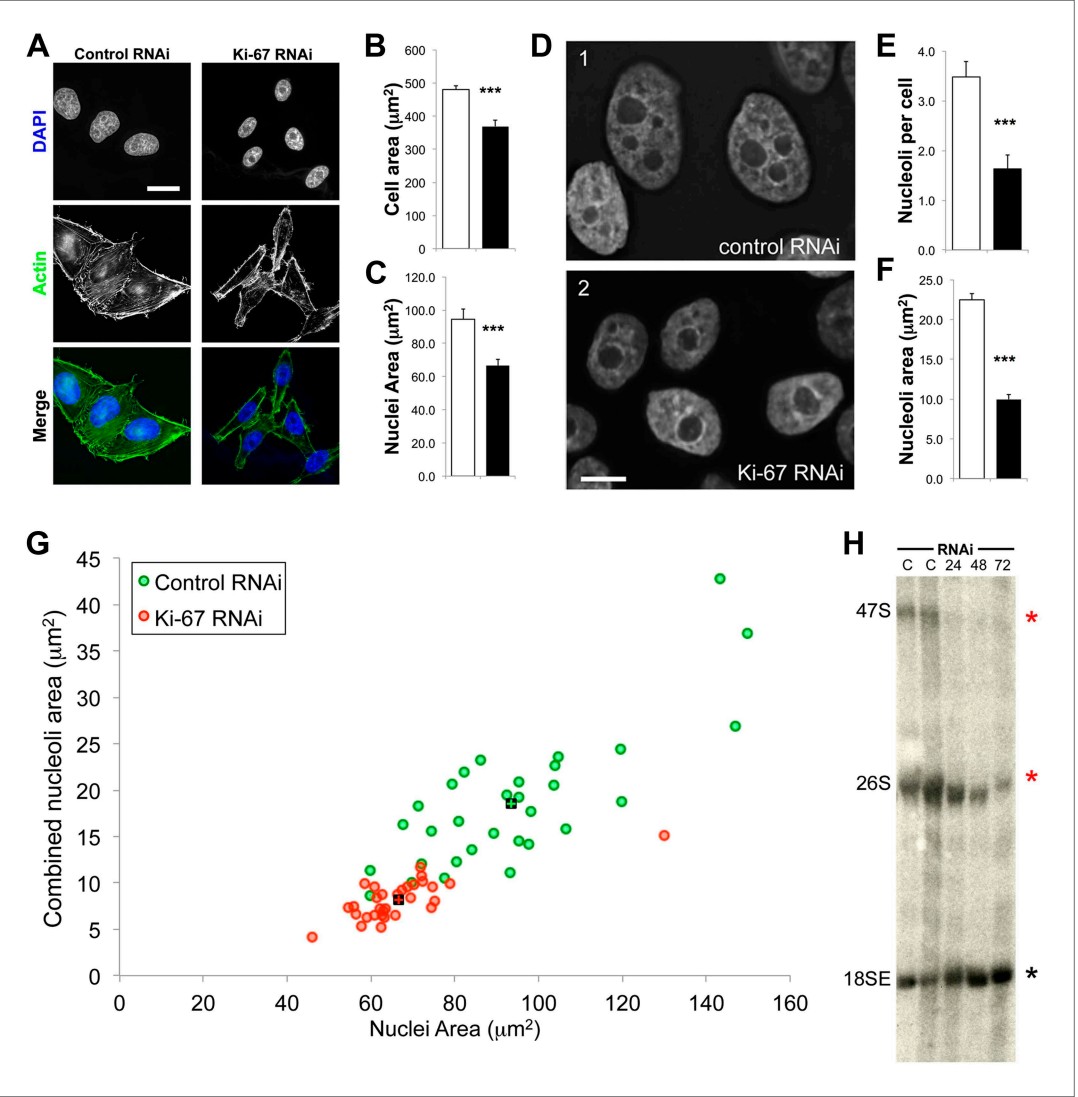

**Figure 7**. The nucleoli of Ki-67 depleted cells are fewer, smaller, and functionally repressed. Normal cell, nucleus, and nucleolus size is compromised following depletion of Ki-67. (**A**) Representative images showing a reduced cell size and nucleolar size phenotype in Ki-67 depleted cells, using Rhodamine Phalloidin and DAPI as markers. Scale bar 5 μm. (**B** and **C**) Quantification of cell area and nuclear area in control (white bar) and Ki-67 depleted (black bar) cells. Bars show mean ± SEM. $n_{cell}$ = 100. (**D**) Representative images showing the small nuclei phenotype, with single nucleoli following depletion of Ki-67. Scale bar 5 μm. (**E** and **F**) Quantification of nucleolar number and area in control (white bar) and Ki-67 depleted (black bar) cells. Bars show mean ± SEM. $n_{cell}$ = 50. (**G**) A 2D scatter plot showing combined nucleolar area (per cell) on the Y axis, vs nuclear area on the X axis, for control (green) and Ki-67 depleted (red) cells. Each individual translucent dot represents one cell. Black squares represent the means. (**H**) Northern blot of RNA samples prepared from control (C) or Ki-67 depleted cells, for 24, 48, and 72 hr of Ki-67 knock-down. Blot shows decreased signal for 47S bands and a time course dependent decrease for 26S signal (red stars). No clear change was seen with bands for 18SE (black star).

The following figure supplements are available for figure 7:

**Figure supplement 1**. Ki-67 depletion influences cell size, nuclei size, and nucleoli size.

**Figure supplement 2**. Ki-67 depletion has only minor effects on the cell cycle detected by FACS.

**Figure supplement 3**. Ki-67 depletion influences nucleolar size.

(*Chubb et al., 2002*). In this cell line the 13p locus (visualized as a GFP spot) tends to localise in the nuclear interior (*Figure 8A1*) where it is usually closely associated with a nucleolus (*Figure 8B1*). A nuclear erosion script developed by Bickmore and Perry (*Croft et al., 1999*) was used to divide the nucleus into 5 concentric rings of equal area in order to score the position of the locus within the nucleus (*Figure 8A2*).

Depletion of Ki-67 had a strong effect on the nuclear localization of chromosome 13p. In cells depleted of Ki-67, the tagged 13p locus tended to move from the nuclear interior towards the nuclear periphery (*Figure 8A3*). This reorganization of chromosome 13p was correlated with a disengagement from the nucleolus (*Figure 8B2,3*). Both phenotypes are consistent with a decreased efficiency of reactivation of the 13p NOR.

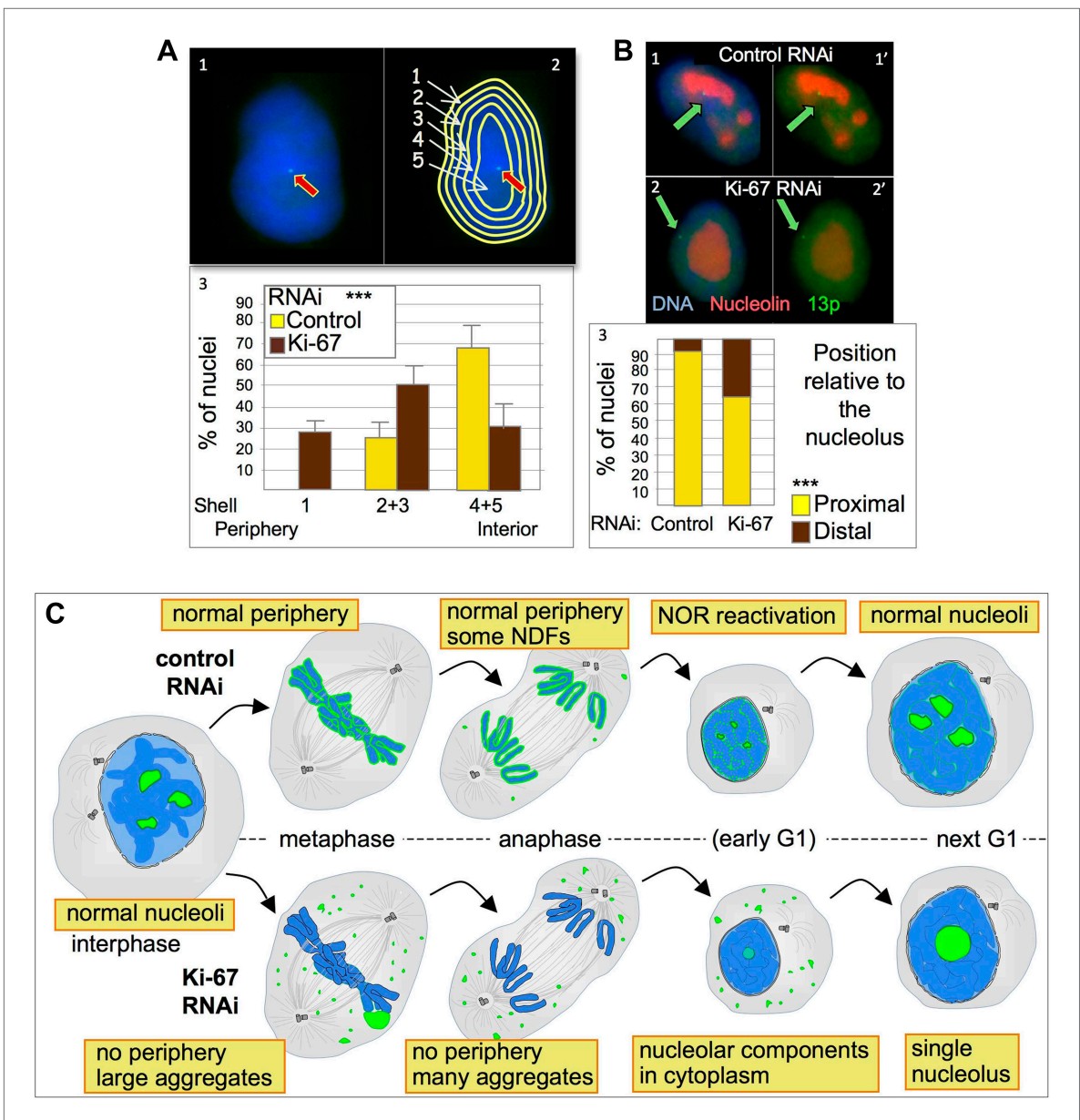

**Figure 8**. Ki-67 depletion affects nuclear architecture. (**A**) The position of a chromosome 13p (marked by a LacO array:LaciGFP) in HT1080 cells (1) was assessed in control and Ki67 RNAi experiments. 150 nuclei from three independent experiments were imaged and analysed with an erosion script software to locate the position of the locus (2). The locus repositions from the interior toward the periphery after Ki-67 RNAi (3). (**B**) The cells described in A were stained for nucleolin after control or Ki-67 RNAi. The association with the nucleolus was recorded as proximal (1–1') or distal (2–2'). (3) Quantification of the analyses. (**C**) Model for Ki-67 function in mitosis; (blue: chromatin; green perichromosomal proteins; grey: microtubules). See text for details.

Thus, although Ki-67 depletion and loss of much or all of the perichromosomal compartment has little obvious effect on chromosome structure or segregation in the first mitosis after depletion, after mitotic exit the nucleus undergoes fundamental changes consistent with decreased activity of the ribosomal gene clusters and with altered interactions between chromosomes. More detailed examination of the link between Ki-67, the perichromosomal compartment, and nucleolar organizer region (NOR) reactivation will be an important subject for future study.

## Discussion

The perichromosomal layer is a little-studied structure whose existence was mentioned in classical studies of mitotic chromosomes (*Schrader, 1944*), but only became widely accepted once its constituents could be described as a result of staining with specific antibodies (*Van Hooser et al., 2005*). Functions proposed for the perichromosomal layer include acting as a protective sheath or pellicle around the chromosomes (*Schrader, 1944*; *Yasuda and Maul, 1990*), serving a role in chromosome condensation or structure (*Gautier et al., 1992c*; *Takagi et al., 1999*; *Kametaka et al., 2002*; *Scholzen et al., 2002*; *Van Hooser et al., 2005*), helping to partition the nucleolar components (*Dundr et al., 2000*; *Van Hooser et al., 2005*), and serving as a platform during nucleolar reassembly (*Savino et al., 2001*; *Hernandez-Verdun, 2011*). All of these hypotheses remained purely speculative, however, as no technique was available to perturb the formation of this layer.

In this study, we have shown that depletion of Ki-67 causes all nucleolar proteins tested to no longer associate with the chromosome periphery during mitosis. Instead they often form enormous aggregates that cap one end of the metaphase plate of aligned chromosomes. Correlative light and electron microscopy confirmed that these 'caps' were neither remnants of partially disassembled nucleoli nor cytoplasmic nucleolus-derived bodies (NDBs) (*Dundr et al., 1996*).

Although it is not practical to obtain specific antibodies to stain for all known components of the perichromosomal layer, we have shown that at least six different proteins fail to localise to the chromosome surface when Ki-67 is depleted. We also confirmed a previous cryoelectron microscopy study (*Gautier et al., 1992b*) and visualized this layer as a dense 'cloud' at the periphery of condensed mitotic chromosomes by thin section electron microscopy. This entire "cloud" surrounding chromosomes is apparently lost when Ki-67 is depleted. Thus, our studies suggest that Ki-67 is required for assembly of many or most components of this layer. It had previously been published that depletion of nucleolin resulted in the loss of both fibrillarin and nucleophosmin (NPM/B23) from the perichromosomal layer (*Ma et al., 2007*). Our studies reveal that nucleolin association with mitotic chromosomes depends on Ki-67, and therefore begin to define a pathway for assembly of the perichromosomal compartment.

How Ki-67 functions in assembly of the perichromosomal layer is not yet determined. We have shown that Ki-67 is a cell-cycle regulated PP1-targeting subunit that is required for regulation of the phosphorylation of nucleophosmin/B23 on S125. Casein kinase II phosphorylation of B23 at S125 is important for regulating its dynamics and molecular chaperone activity (*Szebeni et al., 2003*; *Negi and Olson, 2006*). However, PP1-binding by Ki-67 is not required for its role in assembly of the mitotic chromosome periphery. Ki-67 has been reported to bind DNA (*MacCallum and Hall, 2000a*) and to interact with HP1 family members (*Kametaka et al., 2002*; *Scholzen et al., 2002*). These properties could mediate Ki-67 interaction with the mitotic chromosomes and then provide a platform for the recruitment of other perichromosomal components.

Remarkably, loss of the perichromosomal layer has a dramatic effect on the behaviour of its constituent proteins, but little effect on the mitotic chromosomes themselves during the first mitosis following its loss. After 48 hr of Ki-67 depletion, the mitotic chromosomes appear to be no more susceptible to DNA damage than normal, have a normal morphology in cytogenetic spreads, and behave normally in our intrinsic metaphase structure assay, which can detect defects in chromosome architecture that are not readily apparent by simple examination of the chromosomes (*Hudson et al., 2003*; *Samejima et al., 2012*).

Importantly, nuclei that reassemble following Ki-67 depletion are smaller than control nuclei, the location of at least one chromosomal locus within the nucleus is altered and they possess a single large nucleolus. All of these observations can be explained by two effects of Ki-67 depletion. First, the loss of Ki-67 and the perichromosomal layer apparently allows acrocentric chromosomes to cluster during nuclear reformation with resultant formation of a single nucleolus as a result of fusion of several nucleolus organizing regions (NORs).

Second, the loss of Ki-67 and the perichromosomal compartment also causes a decrease in the efficiency of NOR reactivation during mitotic exit. Thus, the combined nucleolar area of Ki-67-depleted

is >44% less than in control cells, the overall level of rRNA transcription is decreased and the NOR-bearing chromosome 13p frequently loses its association with nucleoli. Although the link between NOR reactivation and association with nucleoli is complex (*Kalmarova et al., 2007*), it is clear that in cells where 13p is distant from nucleoli, its NOR must not be reactivated.

We note that despite having a normal cell-cycle profile at 48 hr, Ki-67 depleted nuclei and cells are also significantly smaller than their control counterparts. This is consistent with components of the translational machinery becoming limiting. For example, mutations in the TOR pathway, an important regulator of protein translation, tend to give rise to smaller cells (*Miron and Sonenberg, 2001*; *Jewell and Guan, 2013*).

Why would NOR reactivation and rDNA transcription be decreased in the absence of Ki-67? Although the nucleolar components we examined exhibit a highly abnormal distribution in mitotic cells following Ki-67 RNAi, all of them ultimately find their way back into nucleoli during the subsequent interphase. Thus, the pathway of nucleolar reassembly exhibits a remarkable degree of plasticity. Furthermore, the segregation of at least one major nucleolar component is noticeably more variable following Ki-67 depletion, suggesting that association with the chromosomes may contribute to the accurate and efficient segregation of the ribosome synthesis machinery at cell division.

In addition to a role in partitioning the protein synthesis machinery, we suggest that association of nucleolar components with the mitotic chromosome periphery is important for NOR reactivation and nucleolar assembly during mitotic exit. During nuclear envelope reassembly, vesicle association and fusion on the chromosomal surface to make the double membrane bilayer precede the assembly of functional nuclear pores (*Kutay and Hetzer, 2008*). Thus, during mitotic exit, there is a brief period during which the chromatin is insulated from contact with the cytoplasm (*Figure 8C*). Their association with the chromosomal surface as the nuclear envelope is laid down at the end of mitosis, means that the ribosomal components of the perichromosomal compartment are inside the membrane barrier when it forms. This early access to NOR chromatin and the high protein concentration within small decondensing G1 nuclei may be critical factors in NOR reactivation, thereby 'jump starting' nucleolar reassembly. Indeed, it has previously been shown that assembly of prenucleolar bodies, an early step in nucleolar reassembly takes place on the chromosome periphery (*Savino et al., 2001*). In the absence of Ki-67 and of the chromosomal periphery there will be a period of time during mitotic exit when nucleolar components in the cytoplasm are denied access to the chromosomes, and this could then explain the decreased efficiency of nucleolar reassembly.

Remarkably, although the abnormalities in the dispersal and re-aggregation of nucleolar components during mitosis appear quite dramatic, Ki-67-depleted cells appear to survive with their single nucleolus and can proceed to another cell cycle. This second division becomes more problematic and we have observed an increase of apoptosis together with mitotic delay. The mitotic defects that occur at the second mitosis after loss of the perichromosomal layer, could be related to a necessary interphase function of Ki-67 or reflect decreased synthesis of a key component required for chromosome segregation, and therefore will require future investigation.

## Perspectives

Our studies have revealed for the first time that Ki-67 functions as a structural/scaffolding protein required for assembly of the perichromosomal compartment on condensed mitotic chromosomes. Thus, Ki-67 has an important role in determining the behaviour of many nucleolar components during mitosis. Interestingly, this method of segregating nucleolar components means that nucleolar reassembly can begin immediately after CDK activity declines and before the re-establishment of nuclear-cytoplasmic transport during mitotic exit. Ki-67 is a PP1-interacting protein, however the biological role of this activity remains elusive and indeed the protein could execute this function during another phase of the cell cycle rather than mitosis. How the various activities of Ki-67 combine to determine the nucleolar morphology in proliferating cells and whether the perichromosomal layer has a subtle role in chromosome dynamics during mitosis or nuclear reformation remain to be determined in future studies.

## Materials and methods

### Cell culture and RNA interference

HeLa Kyoto and MRC5 cells were maintained in DMEM supplemented with 10% and 15% FBS respectively. DT40 cells carrying a single integration of the LacO array (*Vagnarelli et al., 2006*) were cultured in RPMI1640 supplemented with 10% FBS and 1% chicken serum.

For RNAi treatments HeLa cells in exponential growth were seeded in six-well plates with or without polylysine-coated glass coverslips and grown overnight. Transfections were performed using Polyplus jetPRIME (PEQLAB, Southampton, UK) with the indicated siRNA oligos and analysed 48 hr later as previously described (*Vagnarelli et al., 2006*). For the rescue experiments HeLa cells at 50% confluence were transfected with 400 ng of plasmid DNA and 50 nM of siRNA oligonucleotides and analysed 48 hr post-transfection. The siRNA oligonucleotides against Ki67 are as follows:

Ki-1 as published by *Vanneste et al. (2009)*; Ki-2 :5′AAGCACCAGAGACCCUGUATT3′; Ki-5:5′GCA UUUAAGCAACCUGCAA3′; a 21-mer oligonucleotide (CGUACGCGGAAUACUUCGAdTdT) was used as a control (*Elbashir et al., 2001*).

To test for successful depletion of cPerP proteins, HeLa cells were co-transfected with specific siRNA oligos together with the appropriate GFP-cPerP cDNA constructs. Control samples were prepared in parallel via cells transfected with GFP-PerP cDNA only. Following a 48 hr expression/knockdown period cells were harvested and processed routinely for Western analysis. Knockdown was considered successful if Western analysis revealed decreased GFP expression levels in siRNA treated samples (rabbit polyclonal anti-GFP, Invitrogen, Paisley, UK).

## Indirect immunofluorescence and microscopy

The primary antibodies were used as follows: Ki-67 (mouse monoclonal BD Transduction laboratory, Oxford, UK) 1:100; nucleolin (rabbit polyclonal; Abcam) 1:300; NIFK T234ph (Rabbit polyclonal; Abcam, Cambridge, UK) 1:100; Repo-Man (*Vagnarelli et al., 2011*); anti-alpha-tubulin antibody (B512; SIGMA, Gillingham, UK), anti-B23 S125ph, (Abcam); anti-B23T199ph (Abcam); anti-nucleolin (Abcam).

For immunofluorescence, cells were fixed in 4% PFA and processed as previously described (*Vagnarelli et al., 2011*). Fluorescence-labelled secondary antibodies were applied at 1:200 (Jackson ImmunoResearch). 3D data sets were acquired using a cooled CCD camera (CH350; Photometrics) on a wide-field microscope (DeltaVision Spectris; Applied Precision) with a NA 1.4 Plan Apochromat lens. The data sets were deconvolved with softWoRx (Applied Precision). 3D data sets were converted to Quick Projections in softWoRx, exported as TIFF files, and imported into Adobe Photoshop for final presentation.

Live cell imaging was performed with a DeltaVision microscope as previously described (*Vagnarelli et al., 2011*).

For quantification of PP1 binding in vivo, images of prometaphase and interphase transfected cells were acquired and the intensity of PP1 staining at the GFP spot was calculated relative to the average nuclear intensity. The 3D data sets obtained at the same exposure were projected as mean intensities. A 12 × 12 pixel area containing the GFP spot was used to measure the total intensity of the signal. An area of the same size was used to identify the background signal in each cell, and this value was subtracted from the measurement of the nuclear and spot area.

The IMS assay was conducted as previously described (*Hudson et al., 2003*).

For immunoblotting, whole cell lysates were loaded onto polyacrylamide gels. SDS-PAGE and immunoblotting was performed following standard procedures.

## Constructs

Ki-67 (aa 161–659) was obtained by RT-PCR from HeLa cDNA using the following primers: GGATCCGG CGCCACGTTTCCTCTC and CTCGAGTTTTACTACATCTGC and CLONED PGEX4T3 BamHi/XhoI. The PP1-non binding mutant version was generated from this vector using a QuikChange Site-Directed Mutagenesis Kit (Agilent Technologies, Edinburgh, UK). The Lac repressor fusion constructs were obtained by cloning Ki-67 (aa 161–659) into pEGFP:Lac repressor.

hPP1γ was cloned into PET28 EcoRI/HindIII. hNIFK was cloned by RT PCR from HeLa cDNA using the primers CTCGAGGGATGGCGACTTTTTCTGGC and GAATTCTCACTGATTGCTGCTTCT and cloned into the XhoI/EcoR1 sites of pEGFPC1.

The cPERPs were cloned into the gateway system by PCR as previously described (*Ohta et al., 2010*). Accession numbers for cPERPs are as follows: cPERP-A: (C1orf131)–NM_152379.2, cPERP-B: (CCDC137)–NM_199287.2, cPERP-C: (KIAA0020)–NM_014878.4, cPERP-D: (DDX18)–NM_006773.3, cPERP-E: (CIRH1A)–NM_032830.2, cPERP-F: (DDX27)–XM_006723815.1.

## Correlative light and electron microscopy

The CLEM processing method was an adapted version of a previously established protocol (*Booth et al., 2011*, *Booth et al., 2013*). Cells were seeded onto glass-bottomed, gridded dishes (MatTek Corporation, USA) and co-transfected with GFP-cPerpC together with control or Ki-67 specific siRNA

oligos. Following a 48 hr expression period, cells of interest were identified using a wide-field epifluorescence microscope (DeltaVision RT; Applied Precision). GFP-expressing mitotic cells were located and their position mapped using transmitted light to visualise reference coordinates. Cells were then fixed for 1 hr (3% glutaraldehyde, 0.5% paraformaldehyde in 0.2 M sodium cacodylate buffer containing 5 µg/ml Hoechst) and washed in PBS (3 × 5 min). Cells of interest were then re-imaged to acquire micrographs of Hoechst stained chromosomes. Next, cells were osmicated (1% osmium tetroxide in PBS) for 1 hr, washed with PBS (3 × 5 min), ddH$_2$O (2 × 20 min), and then 30% ethanol (1 × 10 min) before contrast staining with uranyl acetate (0.5% in 30% ethanol) for 1 hr. Cells were then dehydrated using a graded series of ethanol washes culminating in 2 × 10 min incubations with 100% ethanol, followed by infiltration with ethanol:resin mixtures (at 2:1 and then 1:1). Finally, cells were embedded in 100% resin, with a gelatin capsule of resin covering the cells of interest, before curing at 60°C for 3 days. Ultra-small resin blocks (50 µm$^2$) were fine trimmed and serial sections (85 nm thickness) taken at areas corresponding to previously chosen coordinate positions, before post-staining in Reynold's lead citrate and uranyl acetate (5% in 50% ethanol) for 10 and 5 min, respectively. Cells were visualised with a Phillips CM120 BioTwin transmission electron microscope (FEI) and micrographs acquired using a Gatan Orius CCD camera (Gatan).

The appropriate Z position correlative light/EM images of a cell were concatenated using ImageJ and then analysed and overlaid using Photoshop Elements 6 (Adobe).

## Chromosome periphery pixel density analysis

Electron micrographs containing metaphase chromosomes from control or Ki-67 depleted cells were used for chromosome periphery analysis. The pixel density of a 1 µm region of interest was measured using the raw data from the 'plot profile' function of imageJ. For unbiased consistency, the 1 µm region of interest always started within the chromosome body and finished in the cytoplasm, with the halfway point lying at the expected origin of the chromosome periphery. Approximately 200 individual pixel density measurements were taken within each 1 µm region. The data were plotted as a line scan profile, with each data point representing the mean of 6 × 1 µm regions of interest per chromosome.

## Scoring cell, nuclei, and nucleolar size/number

HeLa cells were seeded onto coverslips and transfected with control or Ki-67 specific siRNA oligonucleotides. Following a 48 hr knock-down period, cells were fixed using 4% paraformaldehyde, blocked with 3% BSA, and either directly mounted onto slides using hard-setting Vectashield (containing DAPI) or first stained with Rhodamine Phalloidin (Biotium, Inc., Cambridge, UK), before mounting. All measurements were scored using imageJ and area measurement tools. Total cell cross-sectional area was measured using Phalloidin to reveal the cytoplasm and by extension a guide for cell periphery. The area of individual cells or clustered groups of cells were traced and scored using freehand measurement tools.

Nuclei and nucleoli cross-sectional area was measured and nucleolar numbers scored using DAPI as a marker. A contrast threshold was applied to micrographs allowing the semi-automated, unbiased measurement of nucleolar area and number using the ImageJ wand (tracing) tool.

## FACS analysis

Control or Ki-67 depleted HeLa cells were subjected to cell-cycle analysis by FACS. Briefly, 1 × 10$^6$ cells, per condition, were fixed with cold ethanol (70%) for 1 hr, centrifuged, and resuspended in PBS containing RNase A (0.2 mg/ml) and Propidium Iodide (10 µg/ml). Following a 20-min incubation, cells were analysed using FACS. The channel FL2 was used to analyse 20,000 events per condition. Gated cells were manually categorised into cell-cycle stages. Cells were analysed following knock-down periods of 24, 48, 72, and 96 hr.

## Chromosome positioning

HT1080 cell line carrying a LacO integration on chromosome 13p and expressing a Laci:GFP (kindly provided by W Bickmore) was used for depleting Ki-67 in an RNAi experiment as described for HeLa above. At 48 hr the cells were fixed and processed for immunostaining with a nucleolin antibody as described before. 150 nuclei from three independent experiments were imaged and analysed for with the nuclear erosion scrip (*Croft et al., 1999*) to assign the position of the 13p locus.

## Northern analysis

Control or Ki-67 depleted HeLa cells were cultured in 10 cm dishes and processed for Northern analysis. RNA was extracted using fresh TRIzol (Ambion) according to manufacturer's guidelines.

3 μg of RNA per sample was separated using a standard 1.2% TBE agarose gel and transferred to Hybon N+ membrane overnight, by capillary action. Membranes were hybridized in 6X SSPE, 5X Denhardts, 5X SDS at 37°C before washing with 2XSSPE. Probes (see below) were 5' labelled with y32P-ATP and T4PNK.

Probe_d_human AGACGAGAACGCCTGACACGCACGGCAC 5'ETS probe.

Probe_e_human CCTCGCCCTCCGGGCTCCGTTAATGATC 5' end of ITS1 (21S and 18SE).

## Acknowledgements

We thank W Bickmore (Edinburgh) for the HT1080 LacO13p:LaciGPF cell line and W Bickmore and P Perry (Edinburgh) for the nuclear erosion script. This work was funded by The Wellcome Trust, of which WCE is a Principal Research Fellow [grant number 073915]. The Wellcome Trust Centre for Cell Biology is supported by core grant numbers 077707 and 092076, and the work was also supported by Wellcome Trust instrument grant 091020. The work was also supported by a Brunel BRIEF AWARD and the BBSRC grant BB/K017632/1 to PV.

## Additional information

### Competing interests

CPP: Senior Editor, *eLife*. The other authors declare that no competing interests exist.

### Funding

| Funder | Grant reference number | Author |
|---|---|---|
| Wellcome Trust | 073915 | William C Earnshaw |
| Biotechnology and Biological Sciences Research Council | BB/K017632/1 | Paola Vagnarelli |
| Wellcome Trust | Centre Core Grant, 077707 | David Tollervey, William C Earnshaw |
| Wellcome Trust | 077248 | David Tollervey |
| Brunel BRIEF award | LBL301 | Paola Vagnarelli |
| Wellcome Trust | Instrument Grant, 091020 | David Tollervey, William C Earnshaw |
| Medical Research Council Programme Grant | | Chris P Ponting |
| RIKEN Incentive Research Fund | | Masatoshi Takagi |
| Japan Society for the Promotion of Science | KAKENHI, 24657124 | Masatoshi Takagi |
| RIKEN special project funding for basic science in cellular system project research | | Naoko Imamoto |

The funders had no role in study design, data collection and interpretation, or the decision to submit the work for publication.

### Author contributions

DGB, LS-P, CPP, Acquisition of data, Analysis and interpretation of data; MT, KS, (1) Acquisition of data; (2) Revising the article critically for important intellectual content; and (3) Final approval of the version to be published, Acquisition of data, Contributed unpublished essential data or reagents; EP, (1) Acquisition of data; (2) Revising the article critically for important intellectual content; and (3) Final approval of the version to be published, Acquisition of data, Analysis and interpretation of data; GV, (1)Analysis and interpretation of data; 2) Revising the article critically for important intellectual content; and 3) Final approval of the version to be published, Acquisition of data, Analysis and interpretation of data; NI, (1)Conception and design; (2) Revising the article critically for important intellectual content; and (3) Final approval of the version to be published; DT, Conception and design, Analysis and interpretation of data; WCE, Conception and design, Analysis and interpretation of data, Drafting or revising the article; PV, Conception and design, Acquisition of data, Analysis and interpretation of data, Drafting or revising the article, Contributed unpublished essential data or reagents

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
