## [Decision Letter]

[Editors’ note: this article was originally rejected after discussions between the reviewers, but the authors were invited to resubmit after an appeal against the decision.]

Thank you for choosing to send your work entitled “Ki-67 is a PP1-interacting protein that organises the mitotic chromosome periphery” for consideration at *eLife*. Your full submission has been evaluated by a Senior editor, a Reviewing editor (Tony Hyman), and 2 peer reviewers, and the decision was reached after discussions between the reviewers. We regret to inform you that your work will not be considered further for publication at this stage.

As you can see, both reviewers and the editor found the work putting KI67 in a molecular context very interesting, but we felt that as currently presented, there is not enough phenotype data to really understand what is going on, even though we appreciate that you have already put a lot of effort into phenotypic analysis. In general, *eLife* strives to give a clear picture of experiments that are needed for publication, and to ask for experiments that can be completed within a reasonable time frame. The reviewers made a number of suggestions of experiments to strengthen your conclusions, but it seems likely that these will take a significant time to complete, and could not be done within the required time frame. Therefore, we have decided to reject the paper as it currently stands.

*Reviewer*
*#1:*

In this paper, the authors analyze the chromatin-associated protein Ki-67. Ki-67 is the subject of numerous papers in which it is used as a marker for actively dividing cells, but there is very little known about its molecular function. Here the authors demonstrate the Ki-67 is a direct PP1 binding protein in vitro, and that Ki-67 is required for proper PP1 targeting to chromosomes and the dephosphorylation of at least some substrates. In addition, the authors demonstrate that Ki-67 is required for the localization of peripheral chromatin proteins (PERPs) and nucleolar organization. This appears to be a separate function for Ki-67 where it acts apparently as a scaffold rather than a PP1-targeing protein. In general, I found this to be an interesting paper that provides a useful contribution, but there are a number of things that could be done to improve this paper.

1) In the text, there is an inappropriate balance in the amount of space used to describe the different experiments. For example, there are many negative results (i.e., Ki-67 is not involved in something) that receive substantial space. I would suggest that the authors reduce their description of the lack of effect on nucleolin and NIFK phosphorylation to a single sentence. Similarly, they could dramatically reduce their description of the lack of effect on chromatin structure.

2) Instead of focusing on these negative results (even though they are important for framing what Ki-67 does and doesn't do), it would be helpful to have additional information about the key observations that they have focused their title on – the role of Ki-67 in organizing the chromosome periphery and nucleolar organization. Some key functions where the authors establish a role for Ki-67 are less completely explored. The paper is written as if the authors are expecting a key mitotic defect, and then are somewhat surprised that this is not the case. Although this may accurately reflect the nature of the actual experiments, in retrospect it should now be possible to frame a paper that more explicitly analyzes the nature of this new role.

3) In addition to analyzing the potential mitotic roles for Ki-67 (which don't seem that critical based on the data presented in this paper), it would be very helpful to test what else Ki-67 might be doing. For example, the authors show very striking defects in nucleolar organization. Does Ki-67 compromise ribosome biogenesis or growth in some way? What about some aspect of gene expression? Basically, is there a functional consequence to the defective nuclear organization that they identify?

4) The binding assay analyzing the interaction between PP1 and Ki-67 needs to be better controlled. In addition to a positive control for PP1 binding, the authors need to show a negative control (GST alone). There is also not a very complete description of the biochemistry (including the buffers used). I am particularly concerned as PP1 has specific buffer requirements (the presence of metals *–* Maganese) that will affect the behavior of this protein is in their assays.

5) The authors analyze the localization of multiple peripheral chromatin proteins. This is an important addition to the paper. In this case, they tested both known proteins and “novel” proteins that their lab previously identified. Unfortunately, I couldn't find any description of these proteins, including their identity. As this information is central to the paper, they need to provide the ORF or accession numbers for these.

*Reviewer*
*#2:*

In this study Earnshaw and colleagues examine the cell cycle function of the Ki-67 protein that was identified by a monoclonal antibody raised in mice to the nuclei of Hodgkin lymphoma cells. Although little is known about the Ki-67 protein it is found in the nucleus (specifically the nucleolus) of proliferating interphase cells and to a poorly understood region surrounding anaphase chromosomes, which they refer to as the perichromosomal compartment.

The authors note that Ki-67 shares considerable overall homology with RepoMan, a protein phosphatase 1 (PP1) interacting protein and known regulator of chromosome function. Indeed the strongest stretch of homology between the two protein is within a domain of RepoMan that mediates its interaction with PP1. The authors report that Ki-67 is also a PP1 interacting protein but that the cell cycle distribution of RepoMan and Ki-67 are distinct and do not influence each other, suggesting the two proteins control distinct cellular processes. The authors confirm that Ki-67 is located in the nucleolus in interphase and to the perichromosomal compartment in anaphase. Notably, the localization of six other proteins, including cPERP-B, cPERP-C, cPERP-D, cPERP-F, B23 and NIFK (some of which were identified in a previous proteomic analysis of mitotic chromosomes), to the periphery of anaphase chromosomes is severely reduced when the expression of Ki-67 is repressed by RNAi. They suggest from this that Ki-67 may be a scaffold for establishment of the perichromosomal compartment. These results are novel, interesting and believable.

The authors go on to show that the number of nucleoli observed in interphase nuclei is reduced from 3-4 in control cells to only one in cells in which Ki-67 expression is repressed. Disappointingly, however, they find no obvious phenotypic consequence for this effect, although one might expect the 3D architecture of interphase chromosomes to be affected. Rather disarmingly, they also admit to being disappointed with the lack of effect of Ki-67 knockdown on mitotic chromosome structure. Thirdly, they observe no effect on recruitment of other perichromosomal proteins to the periphery of anaphase chromosomes when the association of PP1 to Ki-67 is abolished. Unfortunately, this reviewer is not persuaded by the data in Figure 3 and Figure 3 that Ki-67 controls the phosphorylation status of nucleolin or nucleophosmin/B23. This data set requires considerable further clarification. The reader is left to wonder what the perichromosomal compartment is for, how recruitment of the above mentioned proteins is regulated, and why.

It would seem to this reviewer that one possibility is that the perichromosomal compartment is a vector for the segregation of certain nucleolar components to daughter cells, perhaps akin to the fragmentation of Golgi apparatus during M phase. If this is the case live cell imaging of multiple cells may reveal asymmetric inheritance of nucleolar material. Alternatively, since Ki-67 is exported from nuclei in quiescent cells perhaps Ki-67 is important for nucleolar function on re-entry to the cell cycle? In the absence of such insight this reviewer is unfortunately left somewhat underwhelmed.

---

## [Author Response]

We have performed a number of additional experiments and substantially rewritten the manuscript in order to address all the referee comments. This has resulted in the addition of three new authors, all of whom contributed important experimental results. We thank the referees for giving us useful suggestions that have enabled us to develop a very significant insight into the function of the chromosome periphery in the transition from mitosis to interphase.

In brief, we now provide three independent lines of evidence supporting the novel conclusion that the perichromosomal compartment is required for efficient nucleolar organizer region (NOR) reactivation during mitotic exit. We speculate that by being associated with chromosomes, key nucleolar components are inside the reforming nucleus even before nuclear transport is re-established, thus giving them an opportunity to jump-start nucleolar reactivation. We believe that this is a highly novel and exciting idea that will be tested and expanded upon in many future studies that we believe will be inspired by our present manuscript.

The editorial response before resubmission advised us to focus on two specific aspects:

*“One idea suggested by the reviewers, it would be particularly nice if the authors could conduct additional phenotypic experiments that specifically relate to nucleolar function*.*”*

The revised manuscript includes several experiments showing that Ki-67 and/or the perichromosomal compartment are required for nucleolar reactivation during mitotic exit. For example, we observed decreased rRNA transcription in Ki-67-depleted cells, the aggregate nucleolar size is smaller in those cells, and the localization of the chromosome 13p NOR is altered in interphase nuclei. All of these changes are consistent with a defect in NOR reactivation.

*“It would also be helpful to show that RNAi knockdown of cPERP-B, cPERP-C, cPERP-D, cPERP-F does not influence Ki-67 localisation to the chromosome periphery. If this is the case it would strengthen their argument that Ki-67 is the key scaffold for assembly of the perichromosomal compartment*.*”*

We have conducted the suggested experiment and have shown that the knockdown of 3 other cPERPs does not displace Ki-67 from the chromosome periphery (Figure 4). This suggests that Ki-67 acts upstream during the assembly of this chromosome compartment.

We have now addressed all comments, as described below, and hope that our revised manuscript will engender enthusiasm in both referees and editors alike.

Reviewer #1:

*In this paper, the authors analyze the chromatin-associated protein Ki-67. […] In general, I found this to be an interesting paper that provides a useful contribution, but there are a number of things that could be done to improve this paper*.

We are pleased that the referee felt that this was an interesting paper that provides a useful contribution, and we have attempted to follow the suggestions given for its improvement.

*1) In the text, there is a somewhat inappropriate balance in the amount of space used to describe the different experiments. For example, there are many negative results (i.e., Ki-67 is not involved in something) that receive substantial space. I would suggest that the authors reduce their description of the lack of effect on nucleolin and NIFK phosphorylation (down to a single sentence). Similarly, they could dramatically reduce their description of the lack of effect on chromatin structure*.

We have removed all the negative data on phospho-regulation by Ki-67 as suggested. We have also significantly abbreviated our description of the lack of a role for the perichromosomal compartment in mitotic chromosome structure. We have not totally eliminated the latter, however, as this is the FIRST functional analysis of a chromosomal compartment that comprises some 60 or more proteins and a substantial percentage of the chromosome mass. However we no longer focus on these negative results.

*2) Instead of focusing on these negative results (even though they are important for framing what Ki-67 does and doesn't do), it would be helpful to have much more information about the key observations that they have focused their title on* – *the role of Ki-67 in organizing the chromosome periphery and nucleolar organization. Some key functions where the authors establish a role for Ki-67 are less completely explored. The paper is written as if the authors are expecting a key mitotic defect, and then are somewhat surprised that this is not the case. Although this may accurately reflect the nature of the actual experiments, in retrospect it should now be possible to frame a paper that more explicitly analyzes the nature of this new role*.

This is exactly what we have done, as described below. We have attempted to re-focus the paper so that it is clearly about a functional analysis of the perichromosomal compartment.

*3) In addition to analyzing the potential mitotic roles for Ki-67 (which don't seem that critical based on the data presented in this paper), it would be very helpful to test what else Ki-67 might be doing. For example, the authors show very striking defects in nucleolar organization. Does Ki-67 compromise ribosome biogenesis or growth in some way? What about some aspect of gene expression? Basically, is there a functional consequence to the defective nuclear organization*
*that they identify?*

With the caveat that our paper is about the role of this previously un-studied chromosomal compartment in mitosis, this suggestion turned out to be right on the money. The perichromosomal compartment only exists during mitosis, but it turns out to serve an extremely important function during mitotic exit and early interphase, and we have now developed that story much more completely.

We looked at ribosome biogenesis, and were inspired when we noticed that there is a substantial defect in rRNA transcription. Following on from this we have provided data (all new figures in the revision) that clearly show that not only is ribosomal transcription compromised (Figure 6), the segregation of nucleolar material is more uneven (Figure 5), the aggregate nucleolar size is smaller (Figure 6; Figure 6—figure supplement 1) and the interphase nuclear position of chromosome 13p is altered as its adjacent NOR is frequently not reactivated (Figure 7). Presumably as a consequence of those ribosomal defects, the nuclear and overall cell size of Ki-67-depleted cells is small (Figure. A-C, Figure 6—figure supplement 1). We believe that together, this new evidence strongly supports a key function of Ki-67 (and presumably the chromosomal periphery) during the transition from mitosis to G1.

*4) The binding assay analyzing the interaction between PP1 and Ki-67 needs to be much better controlled. In addition to a positive control for PP1 binding, the authors need to show a negative control (GST alone). There is also not a very complete description of the biochemistry (including the buffers used). I am particularly concerned as PP1 has specific buffer requirements (the presence of metals – Maganese) that will affect the behavior of this protein is in their assays*.

Taking into consideration the quantity of exciting new data inserted in the manuscript we have removed this in vitro study. In fact indications of a possible in vitro binding activity of Ki-67 to PP1 were previously reported (Hendrickx et al, Chem. Biol., 2009 vol. 16 (4) pp. 365-371). We have retained the in vivo tethering and recruitment experiments that clearly show that Ki-67 can recruit PP1 in vivo and also provide information on the cell-cycle regulation of the interaction.

*5) The authors analyze the localization of multiple peripheral chromatin proteins. This is an important addition to the paper. In this case, they tested both known proteins and “novel” proteins that their lab previously identified. Unfortunately, I couldn't find any description of these proteins, including their identity. As this information is central to the paper, they need to provide the ORF or accession numbers for these*.

We have added the accession numbers for those proteins.

Reviewer #2:

*In this study Vagnarelli et al. examine the cell cycle function of the Ki-67 protein that was identified by a monoclonal antibody raised in mice to the nuclei of Hodgkin lymphoma cells. […] They suggest from this that Ki-67 may be a scaffold for establishment of the perichromosomal compartment. These results are novel, interesting and believable*.

We are pleased that the referee finds our data on the requirement for Ki-67 in assembly of the perichromosomal compartment to be convincing.

*The authors go on to show that the number of nucleoli observed in interphase nuclei is reduced from 3-4 in control cells to only one in cells in which Ki-67 expression is repressed. Disappointingly, however, they find no obvious phenotypic consequence for this effect, although one might expect the 3D architecture of interphase chromosomes to be affected*.

We looked more closely at the effects of loss of the chromosome periphery on nucleolar biogenesis and the 3D architecture of interphase chromosomes in the subsequent cell cycle. This yielded very significant insights and a number of new figures for the manuscript as described above.

*Rather disarmingly, they also admit to being disappointed with the lack of effect of Ki-67 knockdown on mitotic chromosome structure. Thirdly, they observe no effect on recruitment of other perichromosomal proteins to the periphery of anaphase chromosomes when the association of PP1 to Ki-67 is abolished. Unfortunately, this reviewer is not persuaded by the data in*
Figure 3
*and*
Figure 3
*that Ki-67 controls the phosphorylation status of nucleolin or nucleophosmin/B23*.

We agree that our excessive presentation of negative data had obscured our positive results. All that remains from those studies is the demonstration that Ki-67 is important for the removal of S125 phosphorylation from B23, which we leave in as support that Ki-67 does actually function in some aspects of phosphoregulation in vivo. The negative results showing, amongst other things, that Ki-67 does *not* influence the phosphorylation status of nucleolin have now been removed from the manuscript. Hopefully in the very much slimmed-down description of the role of Ki-67 in phosphoregulation, the ambiguities have been removed.

*This data set requires considerable further clarification. The reader is left to wonder what the perichromosomal compartment is for, how recruitment of the above mentioned proteins is regulated, and why*.

We feel that as a result of our many additional experiments, we can now propose a novel – and reasonable – hypothesis for the function of the perichromosomal compartment.

*It would seem to this reviewer that one possibility is that the perichromosomal compartment is a vector for the segregation of certain nucleolar components to daughter cells, perhaps akin to the fragmentation of Golgi apparatus during M phase. If this is the case live cell imaging of multiple cells may reveal asymmetric inheritance of nucleolar material. Alternatively, since Ki-67 is exported from nuclei in quiescent cells perhaps Ki-67 is important for nucleolar function*
*on re-entry to the cell cycle?*

These are the key comments in this review. In our opinion, the reviewer is half right in each case, and when those two halves are put together, then an exciting proposal for the role of the perichromosomal compartment emerges.

Yes, the perichromosomal compartment is a vector for the segregation of many key nucleolar components, but we believe that this is not solely so that daughter cells get equal amounts. At the suggestion of the referee we looked at this in live imaging experiments (new Figure 5). In the absence of Ki-67, nucleolin segregation does become slightly more random. So we believe that the chromosomes are vectors, but the key thing is that the perichromosomal compartment is associated with the chromosomes, and therefore *inside* the reforming nuclear envelope.

Concerning nuclear import/export of Ki-67. For us this is the key to our model. We demonstrate clearly that in the absence of Ki-67 (therefore the absence of most or all of the perichromosomal compartment) NOR reactivation is less efficient during mitotic exit and subsequent ribosomal transcription is reduced. This leads us to propose the model stated on the first page above, that having nucleolar components INSIDE the reforming nuclear envelope, before nuclear pores reassemble and the envelope becomes transport-competent allows those components to act – either early, or in a more concentrated environment, or at an appropriate cell cycle stage – more efficiently and promote more efficient NOR reactivation.

We believe that this is a fundamental and exciting hypothesis that will interest many people who study ribosomal transcription and nucleolar assembly. However, given the weight of experiments already presented here and our desire to focus this manuscript on the first ever (and hopefully definitive) characterization of the function of the mitotic chromosome periphery compartment, we argue that tests of the hypothesis should occupy future studies and are beyond the scope of this manuscript.